# The Cell-Mediated Immune Response against *Bovine alphaherpesvirus 1* (BoHV-1) Infection and Vaccination

**DOI:** 10.3390/vaccines11040785

**Published:** 2023-04-02

**Authors:** Cecilia Righi, Giulia Franzoni, Francesco Feliziani, Clinton Jones, Stefano Petrini

**Affiliations:** 1National Reference Centre for Infectious Bovine Rhinotracheitis (IBR), Istituto Zooprofilattico Sperimentale Umbria-Marche “Togo Rosati”, 06126 Perugia, Italy; 2Istituto Zooprofilattico Sperimentale della Sardegna, 07100 Sassari, Italy; 3Department of Veterinary Pathobiology, College of Veterinary Medicine, Oklahoma State University, Stillwater, OK 74078, USA

**Keywords:** BoHV-1, humoral immunity, cell-mediated immunity, infection, vaccines

## Abstract

*Bovine Alphaherpesvirus 1* (BoHV-1) is one of the major respiratory pathogens in cattle worldwide. Infection often leads to a compromised host immune response that contributes to the development of the polymicrobial disease known as “bovine respiratory disease”. After an initial transient phase of immunosuppression, cattle recover from the disease. This is due to the development of both innate and adaptive immune responses. With respect to adaptive immunity, both humoral and cell-mediated immunity are required to control infection. Thus, several BoHV-1 vaccines are designed to trigger both branches of the adaptive immune system. In this review, we summarize the current knowledge on cell-mediated immune responses directed against BoHV-1 infection and vaccination.

## 1. Introduction

*Bovine alphaherpesvirus 1* (BoHV-1) belongs to the genus *Varicellovirus* in the subfamily *Alphaherpesvirinae* within the family *Herpesviridae.* BoHV-1 is an important pathogen of cattle and buffaloes.

The virion (150–200 nm) consists of an outer structure (envelope) consisting of glycoproteins and an internal tegument (matrix) involving the icosahedral capsid that encloses the condensed linear double-stranded DNA molecule (125–241 kbp) that forms the nucleus of the viral particle. The genome contains 70–170 genes.

Subfamily-specific genes are contained in the internal and terminal inverted repeats (Ir_L_/TR_L_ and IR_s_/TR_S_), all covered in the genome unique short sequence (Us).

A poorly defined complex domain within the viral particle is the tegument. On the surface of the tegument, there is an envelope consisting of a lipid bilayer containing different viral glycoproteins [1]. These proteins harbor multiple epitopes recognized by the host’s immune system (humoral and cell-mediated), and some are responsible for binding neutralizing antibodies to the homologous virus or lead to cross-neutralization with heterologous viruses.

Protection against recurrent infections or reinfections is mainly due to the cell-mediated immune response rather than the humoral immune response [1].

Several countries are working to control or eradicate BoHV-1, which is responsible for significant economic losses in the livestock industry worldwide [2,3,4].

BoHV-1 is considered in the latest European (EU) legislation under the so-called “Animal Health Low” and subsequent delegated acts [5,6]. European Member States were given the possibility to control IBR/IPV in cattle and buffalo.

Based on Regulation (EU) 2018/1882, IBR/IPV is in the C + D + E category, and *Bos* ssp. and *Bubalus* ssp. are vulnerable species. In addition, the DIVA (differentiating infected from vaccinated animals) strategy will be used to control and facilitate eradication plans for IBR infection [2,7]. The goal of these strategies is to reduce the risk factors for introducing BoHV-1 into herds. Indeed, the recent literature provides evidence that different risk factors play a role in introducing BoHV-1 into a cattle herd. In fact, recent literature suggests that different risk factors (herd size, purchase of cattle, cattle density, age of cattle, distance to neighboring cattle herds, and professional visitors) are involved in the introduction of BoHV-1 into herds.

However, when eradication is applied, mitigating the risk of those factors should be considered. In addition, the presence of other animals (e.g., sheep) has a negligible risk. Another important aspect of reducing the risk of introducing BoHV-1 into livestock herds is educating and training farmers, veterinarians, and other professional visitors. Cattle must be carefully evaluated during purchasing to reduce the risk of introducing the infection.

An essential aspect of risk reduction is the vaccination of animals, which protects animals from clinical disease. Vaccines, in generally, result in a good humoral and cell-mediated immune response [7,8].

However, protection against a pathogen primarily depends on innate immune and adaptive immunity. The innate immune system, which comprises physical barriers; small bioactive molecules (including complement proteins, defenses, and cytokines); and cells, such as macrophages or dendritic cells (DCs), is designed to detect invading pathogens. In contrast, adaptive immunity is activated later to clear the infection. It is characterized by exquisite specificity for its target antigens but is much slower to respond to pathogens than innate immune response. The main players in adaptive immune response are effector cells, such as B and T cells, and soluble factors (such as cytokines) [9].

In this review, we summarize the relevant literature on the cell-mediated immune response following BoHV-1 infection and vaccination. This review is an integration of our previous report [10] on the passive immunity of calves.

## 2. BoHV-1 Infection

### 2.1. Acute Infection and Clinical Disease in Cattle

Three BoHV-1 subtypes exist; BoHV-1.1 (1), BoHV-1.2a (2a), and BHV-1.2b (2b) have been identified using genomic and antigenic analyses [11]. Subtype 1 strains generally cause infectious bovine rhinotracheitis (IBR) and are frequently detected in the respiratory tract or aborted fetuses. Respiratory forms of BoHV-1 are the most common (subtype 1) strains in diseased cattle in the United States. Fever, coughing, conjunctivitis, inflamed nares, nasal discharge, anorexia, and dyspnea are common clinical symptoms of these respiratory diseases. After the onset of clinical symptoms, recovery usually takes 4–5 days. Life-threatening pneumonia can occur if secondary bacterial infections occur.

Subtype 2a can also be detected in a subset of IBR cases: infectious pustular vulvovaginitis (IPV), balanoposthitis (IPV), and abortions [12]. BoHV-1 subtype 2b strains can also be detected in certain cases of respiratory disease and IPV/IPB but not abortions [12,13]. Generally, subtype 2b strains are detected in Australia or Europe but not Brazil, and these strains are typically non-pathogenic [14].

Bovine respiratory disease (BRD), a polymicrobial disease, is initiated by several respiratory viruses, including BoHV-1 and can lead to bacterial pneumonia. Stress also increases the incidence of BRD. BRD remains the complex of greatest economic importance in the beef/dairy cattle industry. BRD causes approximately 75% of morbidity and >50% of mortality in US feedlot cattle [15,16,17,18,19].

A major bacteria, *Mannheimia haemolytica* (MH) [20], is a normal flora in the upper airways of healthy ruminant animals [21]. However, MH can enter the low respiratory tract during stress or co-infections [22], causing bronchopneumonia in BRD cases [23,24,25,26]. BoHV-1 infection erodes mucosal surfaces and causes lesions in the upper respiratory tract [27,28]. *MH* then migrates to the lower respiratory tract. Co-infection of calves with BoHV-1 and *MH* typically causes life-threatening pneumonia [29]. Furthermore, BoHV-1 acute infection impairs CD8^+^ T cell recognition of infected cells and CD4^+^ T cell functions, reviewed in [30]. Furthermore, the virus is the commonly diagnosed cause of abortion in North American cattle [31]. Reactivation from latency can lead to abortions weeks or months after acute infection. Infection by virulent field strains or vaccination with modified live vaccines can induce abortion storms, which can range from 25% to 60% of cows.

### 2.2. BoHV-1 Latency–Reactivation Cycle

*Bovine alphaherpesvirus 1* (BoHV-1) has a dual life-cycle in cattle: (1) high levels of virus shedding occur during acute infection, and (2) life-long latency is subsequently established and maintained [32]. While acute BoHV-1 infection is generally initiated in the oral, nasal, or ocular cavities (facial area), the genital tract is also a site for acute infection. High levels of virus replication lead to programmed cell death, inflammation, virus shedding, and disease.

Expression of viral genes during productive infection takes place in three phases: immediate early (IE), early (E), and late (L). Viral gene expression is regulated by transcription unit IE 1 (IEtu1), which encodes two proteins (bICP0 and bICP4). The innate immune responses, and these two proteins promote productive infection and virus production [33,34]. IEtu2 encodes the bICP22 protein, and this protein is predicted to enhance virus replication [34]. VP16, a viral tegument protein, interacts with two cellular proteins: host cell factor 1 (HCF-1) and Oct-1. This VP16/HCF-1/Oct-1 complex initiates IE transcription by specifically binding sequences within all IE promoters [35]. Typically, E proteins are non-structural proteins that promote viral DNA replication. L proteins comprise the infectious virus.

Acute infection generally occurs in the oral, ocular, or nasal cavities. Viral particles produced during acute infection then enter the peripheral nervous system via cell–cell spread to sensory neurons in trigeminal ganglia (TG), an important site for latency, reviewed in [36,37]. Generally, low levels of viral lytic cycle gene expression and virus production occur in sensory neurons compared with infected mucosal epithelial cells. Hence, a subset of infected neurons survives, and latency is established. Maintenance of latency is operationally defined as the long-term repression of viral lytic cycle gene expression, and virus production is not detected. In latently infected sensory neurons, the latency-related (LR) gene locus is the only viral gene abundantly expressed [38]. The LR gene encodes multiple products, which include (1) two micro-RNAs, (2) an abundant non-polyA RNA that exhibits a nuclear localization, (3) alternatively spliced mRNA that can be translated into proteins with a common N-terminus derived from ORF2, and (4) a protein encoded by a transcript antisense to the LR-RNA (ORF-E) [36,37]. Collectively, LR gene products support the survival of infected neurons by suppressing cell death and viral gene expression but promoting neuronal differentiation.

BoHV-1 reactivation from latency is consistently initiated by the synthetic corticosteroid dexamethasone (DEX) [36,37]. LR gene expression is nearly undetectable in TG neurons within hours after a single IV DEX injection. During the early stages of DEX-induced reactivation, latently infected calf TG neuronal cells exhibit viral lytic cycle RNA expression [38,39] and T cell persistence in TGs undergoing apoptosis [38]. Within 90 min of DEX treatment of latently infected calves, two viral regulatory proteins, bICP0 and VP16, are readily detected in the same neuron.

Two late viral proteins, glycoprotein C and D, are detected in a small percentage of TG neurons by 6 h after DEX treatment. Interestingly, VP16 is a late gene, suggesting that a novel mechanism inducing rapid VP16 expression is triggered by cellular factors after DEX administration. Since VP16 specifically activates IE promoters [35,40], VP16 expression would trigger IE gene expression and virus production following DEX treatment. Many bICP0+ and VP16+ neurons express the glucocorticoid receptor (GR), suggesting that these TG neurons are more likely to reactivate. The IEtu1 promoter, which drives the expressions of bICP0 and bICP4, is stimulated by DEX and contains two consensus GR binding sites that are essential for GR and DEX-mediated transactivation [41,42]. Hence, bICP0 and bICP4 would likely be expressed in TG neurons that do not express VP16.

Calves latently infected with BoHV-1 consistently contain viral DNA in their pharyngeal tonsils (PTs): however, virus shedding and viral lytic cycle RNA expression are not detected in PT during latency, suggesting that a latent/quiescent infection is established [43]. Interestingly, LR gene expression is not readily detected in PT during latency. In situ hybridization revealed that viral lytic cycle gene expression is readily detected in PT within 6 h after an intravenous DEX injection of latently infected calves but not prior to DEX treatment [43]. A recent study revealed that bICP4 is expressed within 30 min of DEX treatment in the PT of latently infected calves [44]. Studies designed to identify PT cells that harbor viral genomes are underway. Despite BoHV-1 being a neurotropic Alphaherpesvirinae subfamily member, we predict that PT is a biologically relevant site for virus shedding and transmission during reactivation from latency.

Furthermore, when two herpesviruses co-infect an organism, genetic recombination can occur. Petrini and *coll.* [45] described this phenomenon. In addition, recombination in BoHV-1 can impact the choice of vaccine type and its consequences [46,47].

## 3. BoHV-1 Vaccination

There are several commercial BoHV-1 vaccines on the US market that can be classified as either modified live virus (MLV) or whole killed virus [48]. More than 40 years ago, many MLVs were developed on cell cultures. These vaccines evoke a high humoral and cell-mediated immune response because of attenuated virus replication. Current MLVs establish latency and periodically reactivate from latency, which leads to virus shedding and transmission to pregnant cows and can cause abortions [49]. In fact, vaccination with a widely used commercially available MLV reduced live births relative to no vaccination [50]. MLV vaccines are immunosuppressive and can be pathogenic in calves because their immune system is not fully developed. Vaccine outbreaks in feedlot cattle can induce IBR outbreaks [51,52]. A diverse collection of vaccine and virulent field strains has been sequenced, and significant differences were identified; consequently, PCR primers were designed to identify MLV strains versus virulent field strains [53]. This knowledge allows for the identification of vaccine strains or emerging BoHV-1 strains. MLVs currently used in the United States are not DIVA vaccines. Killed vaccines generally comprise whole viral particles that are inactivated by chemical treatments. These vaccines, to be effective, require more than one vaccination to determine a protective neutralizing antibody titer against the challenge. Furthermore, killed vaccines do not consistently induce cellular immune responses.

Within the European Union (EU), DIVA vaccines are required; thus, vaccinated animals versus animals infected with a virulent field strain can be identified. Two types of commercial marker vaccines are currently available. The first contains deletion of the glycoprotein E (gE) gene and the second type contains deletions of gE and the viral thymidine-kinase (TK) gene. The gE gene encodes a protein required for anterograde transport from TG to ocular surfaces and nasal cavities [54]. Hence, if a gE DIVA virus establishes latency, reactivation from latency should not readily occur. The viral TK gene encodes a protein that phosphorylates thymidine and is important for viral replication in non-dividing cells. Although deleting TK in the context of the gE gene will further reduce the chance of reactivation from latency, a thymidine kinase BoHV-1 mutant was reported to reactivate from latency and to cause abortions [55,56]. Hence, merely mutating the TK virus is not sufficient for producing a safe and efficacious BoHV-1 MLV. However, the use of vaccination in the European country has led to Austria, Germany, Denmark, Finland, Sweden, a region in the United Kingdom (Jersey), an area of Italy known as Valle d’Aosta and an Italian Province named Bolzano, and the Czech Republic being considered BoHV-1-free [2,5]. Moreover, in several parts of France there are areas that have started an IBR eradication programme [57].

## 4. Innate Immune Cells: Bridging Innate and Adaptive Immune Responses

BoHV-1 infection induces transient initial immune-suppression; however, animals can develop both innate and adaptive immune responses that clear infection [38,58].

Macrophages and dendritic cells (DCs) are the main players during innate immunity after infection. DCs and macrophages also drive the development of adaptive immune response: because they detect and uptake foreign molecules, and then process and present antigens to T lymphocytes [59,60]. Natural killer (NK) cells are also key innate immune modulators that kill pathogen-infected or malignant cells [61]. A summary of the functions and roles that macrophages, DCs, and NK cells play after infection are summarized below.

### 4.1. Macrophages

Macrophages are the first line of defense against pathogens. It is well established that macrophages are professional phagocytes mediate immune responses to pathogens. Macrophages enter tissue and influence organogenesis, tissue injury, malignancy, sterility, or pathogenic inflammatory stimuli. Subsequently, macrophages are activated to eliminate the cause of their influx and to restore homeostasis. To date, the main studies on macrophages involve the TLR4 agonist lipopolysaccharide [59].

Macrophages possess a vast array of pattern recognition receptors (PRRs), which are designed to detect specific molecular structures located on the surface of pathogens (pathogen-associated molecular patterns, PAMPs), apoptotic cells, and damaged senescent cells. PPRs can produce non-specific anti-infection, anti-tumor, and other immune-protective effects.

To date, based on the protein domain, the following types of PPR are known: (1) Toll-like receptors (TLRs); (2) nucleotide oligomerization domain (NOD)-like receptors (NLRs); (3) retinoic acid-inducible gene-I (RIG-I)-like receptors (RLRs); (4) C-type lectin receptors (CLRs); and (5) absent in melanoma-2 (AIM2)-like receptors (ALRs). The main function of PRRs is to recognize and bind their corresponding ligands; then, adaptor molecules are recruited, determining downstream signal pathways to exert an effect [62].

BoHV-1 can infect alveolar macrophages, but MDBKs produce higher levels of the virus following infection [63,64]. Monocytes are also susceptible to BoHV-1 in vitro [65], but to a much lesser extent than MDBK [66]. BoHV-1 infection of macrophages can impair their major histocompatibility complex (MHC)-mediated antigen processing and presentation. The viral envelope protein UL49.5 interferes with peptide transport for MHC loading, resulting in MHC-I down-regulation in virus-infected cells [67,68]. In addition, protein virion host shutoff (VHS) activity can shut down the synthesis of MHC class I and MHC class II [69,70]. The suppression of MHC I and MHC II antigen presentation reduces T cell stimulation, which is an important BoHV-1 immune evasion strategy [71,72]. BoHV-1 infection also impairs phagocytic activity of macrophages, and their antibody-dependent cellular cytotoxicity function [63,73]. For example, a study reported that alveolar macrophages infected in vitro with BoHV-1 presented reduced Fc-mediated receptor activity and phagocytosis compared with uninfected controls at 12 and 6 h (h) post infection (pi), respectively [63].

Other studies concluded that BoHV-1 infection triggered IFN-α release from alveolar macrophages and monocyte-derived macrophages (moMφ) in vitro [63,74]. Furthermore, it was reported that these cells are primary contributors of type I IFN production early after infection (3–4 days pi) in vivo [73,75]. Macrophages also contribute to the ‘early’ release of TNF and other pro-inflammatory cytokines in vivo, which are involved in controlling infection [58,73,75]. These early cytokine responses influence the ability of the host to overcome a primary BoHV-1 infection, limiting viral replication and promoting the development of other effector immune functions [73,75]. The ability of macrophages to produce cytokines that promote leukocyte chemotaxis in the inflammatory sites and their ability to kill infected cells regulate pathological changes associated with this disease [73]. From 5–10 days post infection (dpi) [71], macrophages are stimulated by IFN-γ released from activated T cells, which enhances their ability to kill virus-infected cells [58,75].

### 4.2. Dendritic Cells

Dendritic cells (DCs) are cells of the innate immune system and professional antigen presenting cells (APC) [60]. DCs are considered sentinels of the immune system and express a broad range of PRRs. For example, DCs can uptake and process proteins and subsequently migrate into secondary lymphoid tissues to present processed antigens to lymphocytes [60,62]. DCs uniquely prime naive T cells [60]. Plasmacytoid DCs (pDCs) are a subset of DCs that release high levels of type I IFN in response to viral infection and to TLR-9 agonists and represent the main source of type I IFN during acute foot and mouth disease virus (FMDV) [76,77].

Despite the pivotal role played by these cells in driving and shaping adaptive immune responses, there are few published studies defining how BoHV-1 influences bovine DC phenotype or functions. However, blood derived DCs are not susceptible to BoHV-1 infection (LAM strain) in vitro, whereas monocytes exhibit low infectivity while MDBK cells are very susceptible to infection [66]. However, DCs (and not monocytes) present viral antigen to T cells: DCs stimulated strong proliferation of Ag-specific T cells when pre-incubated with live or inactivated wild-type BoHV-1. In addition, when viral attachment to the DC surface was inhibited by pre-treating the virus with soluble heparin, researchers observed a dramatic reduction in T cell proliferation [66].

Notably, a DNA-based vaccine encoding a soluble CD40 ligand (CD40L) and the adjuvant Montanide™ GEL01 (GEL01) activates bovine afferent lymph dendritic cells (ALDCs) in vitro, increasing the expression of MHC II and co-stimulatory molecules [78]. Furthermore, bovine ALDCs were activated (assessed by CD40 up-regulation) by a DNA-based vaccine encoding a truncated BoHV-1 glycoprotein D (pCIgD), either alone or encapsulated in a liposomal formulation containing LPS and MANα1-2MAN-PEG-DOPE (pCIgD-Man-L) [79].

Future studies should investigate interactions of BoHV-1 with the major DC subsets, including conventional DC1 (cDC1), conventional DC2 (cDC2), pDCs (pDCs) [76], and DCs from different organs, to better understand the immune-pathogenetic mechanisms triggered by BoHV-1 exposure.

### 4.3. NK Cells

NK cells are key members of innate immunity that attack pathogen-infected and malignant cells by producing immunostimulatory cytokines, such as IFN-γ and TNF (formerly known as TNF-α). They are activated to kill potential target cells based on signals (inhibitory and/or activating) received from ligands on these cells [61].

Previous studies suggested that NK cells are activated during BoHV-1 infection: these cells kill virus-infected cells to contain BoHV-1 spread [75]. It was predicted that MHC I down-regulation triggered by BoHV-1 provides an activator signal for NK cells [58]. This seems to be activated at an early stage of infection that plays an important role in limiting viral replication [73]. Nevertheless, it may also mediate pathological changes associated with this disease [73]. NK cells are also involved in a later stage of virus infection when Th1 responses develop. This activity peaks as late as 7–10 dpi, in part because this activity is induced when Th cells release IFN-γ [71].

## 5. T Cell Responses against BoHV-1 Infection and Vaccination

T cells are the main players in cell-mediated adaptive immune response [7]. In this part of the review, we focus on the impact of BoHV-1 infection on bovine T cell subsets and T cell responses triggered by infection and vaccination.

### 5.1. Bovine T Cell Subsets

Mammalian T cells express a T cell receptor (TCR), which recognizes antigens presented by MHC molecules.

The TCR consists of α, β, γ, and δ (αβ T cells; γδ T cells). αβ T cells can be further subdivided into two major subsets based on the expression of either CD4 or CD8. CD4^+^ T cells (T helper; Th) are responsible for regulating cellular and humoral immune responses, while CD8^+^ T cells (cytotoxic T lymphocytes) are responsible for killing infected or malignant cells [7].

Th cells orchestrate adaptive immune responses. Bovine CD4^+^ T cell subsets include Th1, Th2, Th17, and T regulatory (Treg) (CD4^+^/CD25^+^Foxp3^+^) cells [71,80,81]. Th1 cells are primarily involved in defense against intracellular pathogens, promote cytotoxicity, and trigger IFN-γ production by NK cells and cytotoxic T cells [82]. Conversely, the Th2 response is mainly characterized by antibody production [83]. Nevertheless, it is widely accepted that Th1 cells and Th2 cells represent the poles of a spectral response. In addition, each species has its own peculiarities, and the differences between mice, human, and bovine have been described [84]. Notably, bovine CD4+ T cells were reported to function as parasite-specific bovine Th cells to co-express IL-4 and IFN-γ. In addition, it was described non-restrictive IL-2 and IL-10 expressions to IFN-γ or IL-4-producing Th cells, respectively, and neither the selective ability of IL-4/IL-10 to suppress bovine Th1 cells nor the selective ability of IL-12 to stimulate this subset [84]. However, the development and maintenance of a Th1 IFN-γ response in bovine can control certain intracellular pathogens, such as *Mycobacterium bovis* [85,86].

CD8^+^ T cells, also called cytotoxic T lymphocytes (CTL), recognize antigens presented by MHC class I molecules. Consequently, CD8^+^ T cells kill infected cells and inhibit spread of obligate intracellular pathogens [7].

γδ T cells represent a small percentage (<5%) of peripheral lymphocytes in rodents and humans, whereas they can represent 60% of total circulating T cells in ruminants, including cattle [87]. Bovine γδ T display was predicted to possess NK-like or CTL-like activity. These cells can be activated through TCRs or other receptors, such as NK receptors (NKR) or PRR. In response to invading pathogens, these cells release IFN-γ, perforin, pro-inflammatory IL-17 (a key Th17 cytokine), anti-inflammatory IL-10, and TGF-β in the framework of a regulatory/suppressive activity [88,89,90]. γδ T can be divided into two main subsets based on the presence or absence of Workshop Cluster 1 (WC1), a molecule that may serve as a PRR and TCR co-receptor. WC1^+^ γδ T cells possess immuno-surveillance functions and respond to inflammation because they can migrate from blood, whereas WC1^−^ γδ T cells are localized in the uterus, gut, and mammary gland [90]. Furthermore, a subset of γδ T can also express CD2. Interestingly, the percentage of circulating bovine CD2^−^ γδ T cells decrease with age, whereas there is a similar percentage of CD2^+^ γδ T cells during the first year of life [91]. Surface expression of CD25 and CD44, such as αβ T cells, can be used to monitor activation of this T cell subset [91,92,93].

### 5.2. T Cell Responses during Primary Infection

BoHV-1 can infect αβ T lymphocytes in vitro, primarily CD4^+^ T cells and to a lesser extent CD8^+^ T cells [94]. This study also concluded that BoHV-1 is unable to infect γδ T cells. In calves, BoHV-1 selectively infects CD4^+^ T cells [95], which led to programmed cell death (apoptosis) [94,96,97,98]. CD4^+^ T cell during acute BoHV-1 infection is likely to impair host immune responses [71].

During primary infection, BoHV-1 also avoids host cytotoxic CD8^+^ T cell recognition through the down-regulation of MHC class I surface expression on infected cells [71]. In addition, the viral protein VHS decreases MHC class I and MHC class II presentation, thus reducing CD4^+^ and CD8^+^ T cell activation [69,70,71]. The initial immune-suppression by BoHV-1 allows for vigorous virus replication in the nasal and upper respiratory tract epithelial cells and facilitates replication of other respiratory pathogens [28,99].

After this initial immune-suppressive stage, cattle develop an adaptive cellular immune response, with subsequent recovery from disease and virus shedding not detected. Unfortunately, BoHV-1 latency always occurs, especially in sensory neurons [28]. Responses to BoHV-1 include both Th1 and Th2 cells [75], although there is a skew towards a Th1 response, characterized by IFN-γ release and strong cytotoxic activity against infected cells, which correlated with enhanced control of infection [72]. IFN-γ release by Th cells indeed activates macrophage, NK, and CTL functions [71]. These events occur by 5 dpi and peak between 7 and 10 dpi. Wang and Splitter observed that CD4^+^ T cells presented sustained cytotoxic activity against macrophages infected with BoHV-1 or pulsed with viral polypeptides. CD8+ T cells lysed macrophages pulsed with BoHV-1 polypeptides through a Fas-mediated activation of apoptosis in target cells [100]. Overall, Th1 response strongly correlates with protection, and several methods can measure the development and intensity of Th1 response directed against BoHV-1.

Soluble IFN-γ levels or IFN-γ-releasing cells are used to measure Th1 responses, and circulating levels of this cytokine are frequently monitored in cattle infected with virulent BoHV-1 isolates. As expected, IFN-γ levels in infected animals correlate with decreased clinical symptoms [101,102].

Proliferation assays are also employed to monitor T cell responses against BoHV-1 [103]. Circulating leukocytes from immunized calves were re-stimulated in vitro with an inactivated virus or its proteins (obtained by immune-adsorbent chromatography). The whole virus, tegument protein VP8, or glycoprotein gIV consistently stimulated the proliferation of lymphocytes from BoHV-1-immunized animals [103].

The quantification of diverse T cell subsets in peripheral blood using flow cytometry during BoHV-1 infection [104,105] or in selected tissues using immunohistochemistry (IHC) has been performed [106,107,108]. BoHV-1 infection increases the prevalence of γδ T cells in peripheral blood lymphocytes early after infection [104]. Furthermore, infection with the Iowa strain concluded that there was a reduced number of CD4^+^ or CD8^+^ T cells, whereas γδ T cells levels were the same [105]. Intranasal inoculation of the same strain resulted in decreased levels of γδ T cells and Tregs (Foxp3^+^) in the lungs of infected animals, whereas a slight increase in CD4^+^ or CD8^+^ T cells was detected in these tissues. IHC studies also revealed an increase in IFN-γ-producing lymphocytes in the lungs after infection [106]. BoHV-1 infection reduced γδ T cells and Tregs (Foxp3^+^) levels in the thymus of acutely infected calves [108]. Collectively, these studies indicate that BoHV-1 infection activates adaptive immune responses, with Th1 responses that control infection and clinical disease. Nevertheless, the phenotype of virus-specific IFN-γ-releasing T cells has not been characterized. The IBR research community should utilize the improved veterinary immunological toolbox to define these T cell subsets better.

### 5.3. T Cell Responses during Latency and Reactivation

The virus developed several immune evasion strategies in latently infected sensory neurons [70]. However, it should be noted that other organs such as tonsils or lymph nodes can represent a site for BoHV-1 persistence [109,110].

In latently infected sensory neurons, the virus exists as an extrachromosomal circularized DNA. In latently infected neurons, only the BoHV-1 region containing the LR-RNA is abundantly expressed, with subsequent inhibition of the lytic virus cycle and induction of an anti-apoptotic state of the latently infected cells [70]. As pointed out above, we have demonstrated that there are proteins encoded by the LR gene that are expressed [110,111,112].

Several studies demonstrated that CD8^+^ T-lymphocytes control infection of other human herpesviruses (HSV-1) in sensory ganglia during latency. It was also reported that IFN-γ and IFN-α act as control factors of recurrent herpetic lesions [70,113]. In mice latently infected with HSV-1, neurons expressing viral antigens surrounded by non-neural cells expressing TNF, IL-2, Il-6, IL10, or IFN-γ were detected [32]. Persistent infiltration of lymphoid cells in the sensory ganglia or spinal cord of mice or guinea pigs infected with HSV-2 was also described [32]. In general, different studies demonstrated that the long-term persistence of T cells in sensory ganglia during latency is responsible for the regulation of the latency–reactivation cycle [32,113]. Overall, these studies suggest that cytotoxic T cells are likely to reduce the incidence of BoHV-1 reactivation from latency in sensory neurons [32,113]. However, further studies are required to better characterize these cells. In addition, the specific roles of other bovine T cells subsets, such as CD4^+^ or γδ T cells, in preventing BoHV-1 reactivation from latency were not characterized and could be an area of future studies.

Elevated corticosteroid levels (due to stressful events, such as prolonged transportation) and/or immune suppression can initiate reactivation from latency. BoHV-1 is consistently reactivated during in vivo experiments by administering glucocorticoids, such as DEX [70,114]. Interestingly, after reactivation from latency, researchers observed a decrease in circulating levels of γδ-T cells, but not CD4^+^ or CD8^+^ T cells [109], and subsequently, activation of the humoral immune response.

### 5.4. T Cell Responses during Vaccination and Subsequent Challenge

As stated above, vaccination is the most effective preventive measure to reduce BoHV-1 circulation and the occurrence of outbreaks [71]. Both humoral and cell-mediated immune responses are required to control infection, and inducing robust T cell memory is critical for long-term immunity [43,71]. Inactivated vaccines primarily induce a humoral immune response and relative short-term memory. By contrast, attenuated MLV vaccines elicit a balanced immune response and long-term memory, but vaccination may result in viral shedding [43]. Immunization with diverse types of vaccines, such as gene-deleted and vectored (either viral and DNA) vaccines, triggers humoral and cell-mediated immunity [43].

Th1 responses play a pivotal role in protection against this virus, and not surprisingly, studies focused on the development of BoHV-1 candidate vaccines often measure the duration and intensity of this response [71].

In this section, several methods of measuring Th1 response during vaccination were described, and these findings are summarized in Table 1.

IFN-γ soluble protein levels or IFN-γ-releasing cells were used to monitor the development and intensity of Th1 response following immunization and subsequent challenge with virulent BoHV-1 isolates [114,115,116]. Petrini and *coll*. Evaluated the safety and efficacy of different DNA vaccines to protect calves against BoHV-1. ELISpot was used to monitor the levels of IFN-γ-secreting cells post vaccination with four diverse DNA vaccines and then challenged with the BoHV-1 Cooper strain 90/180 TN. Calves vaccinated with two (pVAX-tgD; pVAX-tgD + pVAX-48CpG) of the tested DNA vaccines developed milder clinical signs, earlier clearance of challenge virus, and a higher number of IFN-γ-secreting cells when compared with non-vaccinated controls [114]. An independent study demonstrated that a three-gene-mutated BoHV-1 vaccine virus or a gE-deleted vaccine virus induced higher IFN-γ serum levels post immunization compared with sham-vaccinated calves. Furthermore, immunized calves contained elevated IFN-γ serum levels compared with sham-infected controls after challenge with the virulent BoHV-1 Cooper strain, which correlated with protection from clinical disease [115]. Dairy calves immunized with a multivalent vaccine were used to collect PBMCs 5 days post vaccination. Cells were re-stimulated in vitro with live or inactivated BoHV-1, and IFN-γ levels quantified in culture supernatants using a commercial ELISA kit. Vaccination increased BoHV-1-specific IFN-γ production, associated with reduced viral shedding and clinical signs [117]. Later, BoHV-1-specific T cell responses were measured following vaccination with multivalent modified live vaccines in beef calves of different ages (2 or 70 days of age). PBMCs were re-stimulated in vitro with an inactivated virus (Cooper strain), resulting in enhanced IFN-γ levels in culture supernatants at 120 hpi [117,118]. An independent study used ELISA to quantify BoHV-1 specific IFN-γ-secreting cells in vaccinated animals. Animals were immunized with a commercial DNA vaccine based on glycoprotein D (pCIgD) formulated with diverse adjuvants, and PBMCs were collected during the animal trial. Cells from immunized animals were re-stimulated in vitro with inactivated BoHV-1, and the 48 hpi IFN-γ levels in culture supernatants were quantified. This study concluded that the adjuvant improved the cellular immune response elicited by the pCIgD vaccine [119].

Flow cytometry was also used to quantify BoHV-1 specific IFN-γ-secreting cells, fluctuation of T cell subsets levels, and expression of activation markers during BoHV-1 vaccination and challenge [91,92,118,120]. In detail, Endsley and *coll.* monitored the expression of the activation marker CD25 (α chain of IL-2 receptor) on diverse T lymphocyte subsets (CD4^+^, CD8^+^, and γδ T cells) in cattle immunized with a BoHV-1 MLV vaccine, able to confer protection to challenge the virulent isolate. PBMCs were collected longitudinally during the experiment and were re-stimulated in vitro with BoHV-1 ISU99. Mock-infected controls were examined. The measurement of T cell subsets (CD4^+^, CD8^+^, and γδ T cells) from immunized animals revealed a significant increase in CD25 expression compared with controls, suggesting that CD25 was a useful marker for evaluating the induction of BoHV-1-specific T lymphocyte subsets following vaccination. In the same study, a depletion of diverse T cell subsets before incubation in vitro with BoHV-1 ISU99 was carried out; the depletion of CD4^+^ T cells resulted in reduced expression of CD25 on CD8^+^ T cells, but not γδ T cells. Conversely, the depletion of CD8^+^ or γδ T cells did not alter CD25 expression in the remaining T lymphocyte subsets. This indicates that CD4^+^ T cells play an important role in cell-mediated immune responses against BoHV-1 [92].

An independent study monitored the expression of another activation marker (CD44) on circulating γδ T cells in calves immunized with a commercial polyvalent viral vaccine (Vira Shield^®^ 5). The expressions of CD62L and CD45R0, two surface markers used to define the central or effector memory function of bovine CD4^+^ T cells [124], were also investigated. A significant decrease in the percentage of CD2^−^/CD62L^+^ but not CD2^+^/CD62L^+^ γδ T cells after vaccination with Vira Shield^®^ 5 was observed. Furthermore, a decrease in the percentage of CD2^−^ and CD2^+^ γδ T cells expressing CD44 and CD45R was observed [91]. These data suggest that γδ T cells shifted towards a memory phenotype during the in vivo trial. In 2006, the expression of the activation marker CD25 and intracellular IFN-γ levels was measured in diverse T cell subsets (CD4^+^, CD8^+^, and γδ T cells) of cattle immunized with a modified live virus vaccine (containing five major bovine respiratory disease viruses, including BoHV-1). This vaccine conferred protection to challenge virulent BoHV-1 25 weeks post immunization. PBMCs from immunized calves and controls were collected during experiments and then re-stimulated in vitro with BoHV-1. The authors observed that vaccinated animals presented a higher CD25 index versus controls and a higher percentage of virus-specific IFN-γ^+^ T cells after a challenge [120]. All T cell subsets expressed virus-specific IFN-γ, but with higher percentages for CD4^+^ and γδ T cells compared with CD8^+^ T cells [120]. The expression of CD25 was used to evaluate BoHV-1-specific T cell responses following vaccination with a multivalent modified live vaccine in beef calves of different ages (2 or 70 days of age). PBMCs were re-stimulated in vitro by the whole virus, resulting in enhanced CD25 expression, regardless of the age of the immunized calves [118].

Other techniques, such as proliferation assays or cytotoxic T test, were employed to monitor T cell responses in BoHV-1 vaccine trials [116,121,122]. PBMCs were collected longitudinally from calves intranasally infected with the Lam strain or its deleted mutants (gC, gG, gE, and gI) and subsequently challenged with the virulent Iowa strain. This approach revealed higher proliferative responses in calves immunized with gI, gE, and gC mutants compared with controls [121]. Furthermore, calves immunized with 100 ug of affinity-purified BoHV-1 protein VP8 revealed that the PBMCs of these immunized animals presented a high proliferative response when re-stimulated in vitro with VP8 or inactivated BoHV-1 [123]. Formulations containing the adjuvant Montanide™ 1113101PR could improve the cellular immune response, assessed by the virus-specific PBMC proliferative response and release of BoHV-1 IFN-γ in vitro, which correlated with diminishing viral excretion in vivo [116]. Lymphoproliferation assays monitored the ability of a commercial inactivated vaccine to protect cattle against BoHV-1 challenge (Los Angeles strain). At 12 days post challenge (dpc), PBMCs from immunized animals presented a higher proliferation index than controls in response to stimulation in vitro with inactivated BoHV-1, which correlated with lower viral excretion and viral score in vivo [122]. In the same study, the ability of PBMCs to release virus-specific cytokines was assessed: cells were collected 12 dpc (day post challenge) and were stimulated in vitro with inactivated BoHV-1, and immunized animals presented higher levels of IFN-γ, TNF, and IL-4 in culture supernatants compared with controls. This study suggested that the tested vaccine could exert an early adaptive immune response characterized by Th1 and Th2 responses [122]. Nevertheless, the phenotype and functionality of activated T cells was not characterized in detail and should be an area of future studies.

## 6. Systemic and Local Cytokines during BoHV-1 Infection and Vaccination

Cytokines are small proteins that play an essential role in controlling the homoeostasis of the immune system [83]. These molecules frequently act as protectors against both intrinsic and extrinsic dangerous stimuli, such as invading pathogens. As stated above, IFN-γ levels are associated with protection against intracellular pathogens, such as BoHV-1. Nevertheless, an exaggerated immune system activation (associated with an excessive cytokine release) can threaten the host [125,126]. Uncontrolled cytokine release, also known as a “cytokine storm” or “cytokine release syndrome”, is frequently associated with the pathogenesis of several viral diseases, including the SARS-CoV-2 pandemic [125,127]. Several studies demonstrated that BoHV-1 infection resulted in the dysregulation of the cytokine profile in vivo, but often that led to developing a protective immune response. It was reported that this virus induced the release of several cytokines, which inhibited virus replication directly or indirectly by activating effector cells [75]. The changes in cytokine levels associated with BoHV-1 infection are summarized below.

### 6.1. Pro-Inflammatory Cytokines

The most important pro-inflammatory cytokines of the innate immune response are IL-1, IL-6, and TNF [83]. IL-1α and IL-1β are released in the early stages of infection.

These interleukins promote the release of several chemokines, which in turn enhance infiltration of neutrophils and monocytes in the inflamed tissue [128]. IL-6 is a pleiotropic cytokine, which can exert both pro-inflammatory and anti-inflammatory properties.

During infection, this cytokine promotes monocyte recruitment to the site of inflammation and maintains Th17 cells [129]. Tumor necrosis factor (TNF; formerly known as TNF-α) belongs to the TNF superfamily and is involved in host defense against pathogens [130]. It is characterized by a strong pro-inflammatory action; activates vascular endothelium; and triggers the release of several cytokines and chemokines, which recruit leukocytes to the inflammatory site [131]. IL-1β, IL-6, and TNF are potent pyrogens; they promote the synthesis of acute phase proteins from the liver, such as C reactive protein, during inflammation and act on the central nervous system to induce fever and prostaglandin secretion [83,132].

Changes in the levels of these cytokines during BoHV-1 infection were monitored in several studies (see Table 2). Infection with a virulent BoHV-1 isolate (Iowa strain) elicited increased levels of pro-inflammatory TNF, but not IL-1β [101]. Increased levels of TNF were also detected after intranasal administration with either virulent BoHV-1 (IBRV strain HB06) or its attenuated deleted mutants (BoHV-1 gG−/tk−, BoHV-1 gE−/tk−): however, higher TNF levels were detected after infection with the wild-type strain [102]. Only the wild-type strain caused clinical signs. Other studies investigated the effect of BoHV-1 infection on the expression of key pro-inflammatory cytokines in selected tissues. Intranasal administration of the virulent Iowa strain resulted in increased levels of IL-1α and TNF in the lungs, especially in the peribronchial area. Enhanced TNF and IL-1α levels were detected in septal macrophages early post infection. IHC studies revealed that the number of pulmonary alveolar macrophages (PAM) was positive for the tested cytokines, but at lower levels than septal macrophages during the study (1–14 days pi), with only a small increase in TNF at 14 dpi and IL-1α at 1–2 dpi in PAM [107]. Increased TNF levels were observed in the tracheal epithelium of acutely infected calves early pi (6 dpi) but not after DEX-induced reactivation [133]. Increased expression of this cytokine (TNF) was also observed in the central nervous system (frontal and posterior cortex) of acutely infected calves, during reactivation (posterior cortex), but not during latency [134].

Collectively, current studies concluded that the release of pro-inflammatory cytokines early post infection (‘early cytokines’) is predicted to contain viral infection and to promote infiltration of immune cells to the inflamed tissue [58,75,133].

### 6.2. Anti-Inflammatory Cytokines

IL-10 exhibits strong anti-inflammatory and immune-suppressive activities [135,136]. It also counteracts the production and release of pro-inflammatory cytokines and impairs antigen presentation by macrophages and DCs [135,136]. A few studies examined the impact that BoHV-1 has on this immunosuppressive cytokine (Table 3). Changes in circulating IL-10 levels were detected during BoHV-1 infection. Furthermore, infection with a virulent BoHV-1 isolate triggered increased serum levels of IL-10, which paralleled the peak of TNF [101]. IHC was employed to estimate IL-10 levels in the lungs of calves during the intranasal inoculation of the Iowa strain. Infection decreased IL-10 levels in interstitial macrophages: however, enhanced IL-10 levels in lymphocytes from the lung were observed [106]. Future studies designed to provide insight into the role of this cytokine in BoHV-1 immunopathogenesis need to be performed.

### 6.3. Pro Th1 and Pro Th2 Cytokines

In mammals, IL-12 and IL-18 are key inducers of cell-mediated immunity via stimulation of Th1 cells; for example, they enhance cytotoxicity and IFN-γ production by NK cells and cytotoxic T cells [82,137]. IL-4 and IL-13 are critical for induction and perpetuation of the Th2 response, which culminates in antibody production [83]. Nevertheless, some peculiarities in bovine were described: for example, IL-2 and IL-10 expression were not restricted to IFN-γ or IL-4-producing Th cells, respectively. In addition, IL-4/IL-10 and IL-12 do not suppress or stimulate Th1 cells [84]. However, a Th1 IFN-γ response in bovine generally correlates with better control of intracellular pathogens [81].

Infection with a virulent BoHV-1 isolate triggered increased serum levels of IL-12 but not IL-4, which correlated with increased IFN-γ serum levels and decreased clinical signs. IHC was employed to measure IL-12 levels in the lung of calves infected intranasally with the Iowa strain. Infection resulted in increased IL-12 levels in interstitial macrophages early pi and then returned to basal levels at later pi (7 and 14 dpi) [106]. Overall, these data suggest that BoHV-1 infection induces IL-12 synthesis and release, which promotes the development of a protective Th1 response [101,106], (Table 3).

### 6.4. Interferon

Interferons (IFNs) are a family of cytokines that coordinate the immune system in response to viral infections [138]. These proteins were first discovered in 1957 by Isaacs and Lindenmann [139], and three distinct classes of IFNs are known: Type I, Type II, and Type III IFNs [140].

#### 6.4.1. Type I IFN

Type I IFNs are a heterogeneous family with several closely related subtypes. In mammals, eight type I IFN subclasses were described: IFN-α, IFN-β, IFN-Δ, IFN-ε, IFN-κ, IFN-τ, IFN-ω, and INF-ζ [140]. Type I IFNs are the first line of defense in cells during viral infections: after binding to their receptor (IFNAR), these molecules trigger transcription of several genes involved in either immune regulation or with antiviral activity; these genes are called IFN-stimulated genes (ISGs) [141]. Considering the pivotal role of type I IFN following viral infections, it is not surprising that BoHV-1 encodes several genes to inhibit IFN induction and release [30,111].

BoHV-1 infection triggers type I IFN production, which reached peak levels in nasal secretion and blood early pi (36–72 hpi) [58,75]. Recent studies revealed that increased levels of type I IFN (IFN-β) were observed early pi in the serum of calves infected with the virulent (IBRV strain HB06) or its attenuated deleted mutants (BoHV-1 gG−/tk−, BoHV-1 gE−/tk−) [100]. Increased IFN-α and IFN-β (soluble proteins) levels were detected at 3 dpi in the nasal excretion of BoHV-1-infected animals and increased the expression of certain ISGs, including Mx1, OAS, and BST-2, in nasal turbinate or tracheal tissue samples [142]. Increased IFN-β expression was also detected in the nasal mucosa, but not lymphoid tissue, of calves acutely infected with BoHV-1 (Cooper strain) [133]. The same group detected increased expression of IFN-β in the cervical medulla of acutely infected calves, during latency in the cervical medulla and during reactivation in the posterior cortex and cervical medulla [134]. These studies predict type I IFN release, and pro-inflammatory cytokines restrain viral infection and promote early BoHV-1 clearance [75] (Table 4).

#### 6.4.2. Type II IFN

In cattle and humans, there is only one type II IFN: IFN-γ. This cytokine is primarily secreted by T cells and NK cells and possesses strong antiviral activity. It exerts a vast array of immune functions, including inflammation, NK cell activity, and macrophage activation [143,144]. Changes in soluble IFN-γ levels during BoHV-1 infection were monitored in several studies using ELISA tests to characterize a Th1 response (s 4) [101,102,115].

As summarized above (Section 5.2, Section 5.3 and Section 5.4), infection with virulent BoHV-1 isolates (Iowa, IBRV HB06) triggered increased serum levels of IFN-γ [101,102], as well as attenuated BoHV-1 candidate vaccines (tmv; a three-gene-mutated BoHV-1 vaccine virus), BoHV-1 gEdel (a gE-deleted vaccine virus) [115], IBRV HB06-derived mutant BoHV-1 gG−/tk−, and BoHV-1 gE−/tk− [102]. Increased values of type II IFN (soluble proteins) were also detected in nasal excretions in BoHV-1-infected animals, which paralleled viral shedding [142].

Other studies measured IFN-γ-producing cells during BoHV-1 vaccination and/or infection using an IFN-γ ELISPOT (see Section 5.2 and Section 5.3). In summary, a higher number of virus-specific IFN-γ-secreting cells (evaluated through ELISpot) correlated with partial protection conferred by two DNA vaccines. In general, immunized animals developed milder clinical signs post challenge compared with controls: furthermore, earlier clearance of challenge virus was observed (Cooper strain 90/180 TN of BoHV-1) [114].

The ELISA levels of virus-specific IFN-γ-producing cells were measured during BoHV-1 infection or vaccination. PBMCs from immunized animals were collected and re-stimulated in vitro with BoHV-1; after incubation, the values of this cytokine in culture supernatants were quantified by ELISA [116,119,122]. In general, reduced viral shedding (compared with un-vaccinated controls) positively correlated with the increased release of virus-specific IFN-γ detected in PBMCs from immunized calves [116,122].

#### 6.4.3. Type III IFN

IFN-λ1 (IL-29), IFN-λ2 (IL-28A), IFN-λ3 (IL-28B), and IFN-λ4 have been identified in humans and pigs. Epithelial cells are the main source of type III IFNs, whereas macrophages and dendritic cells can also produce these cytokines, but they mainly produce type I IFNs. Type III IFNs, or IFN-λ, are characterized by remarkable antiviral activity and are also important in antiviral protection of epithelial barriers [145]. The effects of humans IFNs are most evident in epithelial cells [146]. In cattle, a type III IFN family member, boIFN-λ3, was cloned and has potent antiviral activity against foot and mouth disease virus (FMDV) in vitro. This study suggested that boIFN-λ3 has potential application as a biotherapeutic to inhibit FMDV and other bovine viruses [147].

IFN-λ3 levels in the nervous tissues (trigeminal ganglion, frontal cortex, posterior cortex, olfactory cortex, and cervical medulla) were examined in calves infected with BoHV-1 (Cooper strain) [148]. Increased IFN-λ3 levels were detected in the TG of acutely infected calves. Interestingly, the other examined nervous tissues did not exhibit increased IFN-λ3 levels. During latency, IFN-λ3 RNA levels were significantly increased in all nervous tissues examined compared with un-infected control calves. During reactivation from latency, IFN-λ3 expression in the frontal and posterior cortex and cervical medulla of BoHV-1-infected calves were down-regulated relative to the control group [148]. Future studies should examine the role of type III IFN during BoHV-1 infection to better understand its antiviral activity and its potential application as a biotherapeutic agent.

## 7. Conclusions

This review summarized the current knowledge on cell-mediated immunity against BoHV-1 infection. The development of an effective cell-mediated immune response occurs despite initial immune-suppression, resulting in impaired MHC-mediated antigen processing and presentation to T cells and apoptosis of infected CD4^+^ T cells. Consequently, clinical disease subsides, and infectious virus is cleared. However, BoHV-1 latency is established, primarily in sensory neurons.

Acute infection leads to the release of type I IFN and pro-inflammatory cytokines, and then, diverse branches of adaptive immunity are activated. Th1 responses are crucial for suppressing viral production. Increasing levels of virus-specific IFN-γ-secreting cells positively correlate with protection and are frequently monitored in experimental trials to test vaccine candidates. Available vaccines do not prevent infection and thus establish BoHV-1 latency.

Understanding the cellular and molecular aspects of immune responses to BoHV-1 infection requires further dedicated research. Future studies should define the phenotype of T cell subsets involved in clinical protection and should clarify the role of the immune system in controlling reactivation from latency. Furthermore, the interaction of BoHV-1 with key immune cells, such as DCs, NK cells, or γδ T cells, should be studied. Finally, a better understanding of the cellular and humoral response of the host–virus interaction should be pursued with the goal of developing safe and effective prophylaxis against BoHV-1. A well-established veterinary immunology toolkit will be essential for achieving these important goals.

## Figures and Tables

**Table 1 vaccines-11-00785-t001:** Th1 response during vaccination against BoHV-1.

Candidate Vaccine	Dose and Route of Vaccine	IBR Strain Used for Challenge	Dose and Route of Challenge Strain	Time Points Tested	Assay Used	Relation to Protection	Reference
pVAX-tgD; pVAX-tgD with pVAX-48CpG; pVAX-UbiLacI-tgD-L; pVAX-UbiLacI-tgD-L with pVAX-48CpG	1 mg of DNA vaccines; IM route	Cooper strain 90/180 TN	5 mL of 10^9.50^ TCID_50_, IN route	0, 13, 21, 28 dpc	IFN-γ ELISPOT	BoHV-1-specific IFN-γ-secreting cells in vaccinated calves; partial protection from virulent BoHV-1 challenge infection	[114]
BoHV-1 tmv (triple gene mutant); BoHV-1 gEdel	1 × 10^7^ PFUs/mL/nostril; IN route	Cooper strain (Colorado-1)	4 × 10^7^ PFUs (2 × 10^7^ PFUs/mL/ nostril); IN route	0, 7, 14 dpc	IFN-γ ELISA (serum)	After challenge, vaccinated calves presented higher IFN-γ serum levels, correlated to lower clinical score than controls	[115]
pCIgD alone or adjuvanted with ESSAI 903110 or Montanide™ 1113101PR	500 μg/dose DNA vaccine; ID route. Booster at day 20 and 33.	Los Angeles strain	3 × 10^6^ TCID_50_/mL; IN route	44 dpv	Proliferation assays, ELISA to measure IFN-γ released by PBMCs restimulated in vitro	Cellular response improved by the addition of adjuvant Montanide™ 1113101PR to the gD vaccine, related to lower clinical score and diminishing viral excretion post challenge	[116]
Commercially available multivalent vaccines containing either modified-live (MLV) bovine herpesvirus-1 (BHV-1) (Bovishield^®^) or MLV plus killed (MLV + K) BHV-1 (Reliant^®^ Plus)	2 mL; SC route.	Cooper strain	4 mL of 10^6.6^ TCID_50_/mL; IN route (2 mL into each naris)	−5, −3, 0, 3, 5, 9, 14 dpc	ELISA to measure IFN-γ released by PBMCs restimulated in vitro with BoHV-1 (inactive or live virus)	Vaccination increased BoHV-1-specific IFN-γ production, associated with a reduction in viral shedding and clinical signs	[117]
Multivalent modified live vaccines	2 mL; IN or SC route. Booster at weaning.	-	-	At median age 2, 70, 140, 217, and 262 days	Proliferation assay, ELISA to measure IFN-γ released by PBMCs restimulated in vitro with BoHV-1	Vaccination increased BoHV-1-specific IFN-γ production	[118]
pCIgD alone or adjuvanted with Montanide™ GEL 01 PR and Montanide™ Essai 903110	500 μg/dose DNA vaccine; ID route.	-	-	21 dpv	ELISA to measure IFN-γ released by PBMCs restimulated in vitro	The group 110 showed the highest IFN-γ secretion levels, suggesting that the adjuvant improved cellular immune response elicited by pCIgD	[119]
Pentavalent modified live virus vaccine containing BVDV type 1, BVDV type 2, BHV-1 (strain Baker), BRSV, and PI-3.	2 mL, SC route	Cooper strain	4 mL of 10^8^ TCID_50_/mL; IN route (2 mL into each naris)	0 dpv, weeks 4, 5, 6, 8, 24, and 25 after vaccination; weeks 1 and 2 after challenge	Flow cytometry (CD25 expression + IFN-γ intracellular levels) of PBMCs stimulated in vitro	Vaccinated animals presented a higher CD25 index and higher percentages of virus-specific IFN-γ^+^ T cells compared with controls. These parameters were correlated with protection to challenge with virulent BoHV-1	[120]
Lam derived deleted mutants (gC-, gG-, gE-, gI-)	2 × 10^4.7^ TCID_50_/mL; IN route	Iowa strain	2 × 10^6.7^ TCID_50_/mL; IN route	14 dpv, 14 dpc	Proliferation assays: PBMCs restimulated in vitro with UV-inactivated purified virus (Lam strain) or gE protein	Higher proliferative responses of PBMCs of calves immunized with gI, gE, and gC mutants compared with controls, related to protection to challenge with virulent BoHV-1	[121]
Affinity-purified BoHV-1 protein VP8	100 μg; IM route. Two booster immunizations at 3-week intervals.	-	-	Before immunization (day 0) and 7 days after each immunization.	Proliferation assays: PBMCs restimulated in vitro with inactivated virus or VP8	Higher proliferative responses of PBMCs of immunized calves compared with controls, indicating that VP8 is a good immunogen	[122]
A commercial inactivated vaccine (Biopoligen Air ^®^)	3 mL, SC route. Booster after 30 days.	Los Angeles strain	10 mL of 1 × 10^6.81^ TCID_50_/mL of BoHV-1; IN route (5 mL into each naris)	12 dpc	Proliferation assays, ELISA to measure cytokines (IFN-γ, TNF, and IL-4) released by PBMCs stimulated in vitro with inactivated BoHV-1	Higher proliferation index and higher release of tested cytokines by PBMCs of vaccinated animals compared with controls, concomitant with lower viral excretion and lower viral score.	[123]
A commercial polyvalent viral vaccine (Vira Shield^®^ 5), containing inhactivated BoHV-1	Booster after 50 weeks.	-	-	5, 3, and 1 weeks pre-vaccination; 1, 3, 7, 11, 25, 47, 50, and 51 weeks post vaccination	Flow cytometry of PBMCs (expression of activation markers on T cell subsets)	Data suggested that γδ T cells shifted towards a memory phenotype after vaccination	[91]
Commercial viral vaccine (Titanium ^®^ 5 L5), containing modified live BHV-1, BVDV1, BVDV2, BRSV, PI3, and killed Leptospira bacterin	2 mL, IM route	Cooper strain	10^8.2^ TCID_50_; IN route	0, 21, 35, 60, and 90 dpv	Flow cytometry (CD25 expression) of PBMCs restimulated in vitro with BoHV-1 ISU99	Vaccinated animals presented a higher CD25 index compared with controls. These parameters were correlated to protection to challenge with virulent BoHV-1	[92]

dpc: day post challenge; dpv: day post vaccination; IM: intramuscolar; IN: intranasal; ID: intradermal; SC: subcutaneous.

**Table 2 vaccines-11-00785-t002:** Modulation of levels of pro-inflammatory cytokines in bovine infected with BoHV-1.

Cytokine	IBR Strain	Virulence of the Strain	Dose and Route of Infection	Post- Infection Time Analyzed	Impact on Cytokine’s Values	Serum/ Tissue	Reference
IL-1α	Iowa	Virulent	2 × 10^7^ TCID_50_, IN route	1, 3, 5, 7, 14 dpi	Increase, mainly in peribronchial area	Lung	[107]
IL-1β	Iowa	Virulent	2 × 10^7^ TCID_50_, IN route	0, 3, 6, 9, 12, 15, 18 and 21 hpi; 1, 2, 4, 5, 7, 9 and 14 dpi	None	Blood (serum levels)	[101]
TNF	IBRV HB06	Virulent	4 × 10^5^ or 4 × 10^6^ or 4 × 10^7^ PFU; IN route	1, 2, 3, 5, 7 dpi	Increase	Blood (serum levels)	[102]
TNF	BoHV-1 gG−/tk−	Attenuated (IBR derived mutant)	4 × 10^5^ or 4 × 10^6^ or 4 × 10^7^ PFU; IN route	1, 2, 3, 5, 7 dpi	Increase	Blood (serum levels)	[102]
TNF	BoHV-1 gE−/tk−	Attenuated (IBR derived mutant)	4 × 10^5^ or 4 × 10^6^ or 4 × 10^7^ PFU; IN route	1, 2, 3, 5, 7 dpi	Increase	Blood (serum levels)	[102]
TNF	Iowa	Virulent	2 × 10^7^ TCID_50_, IN route	0, 3, 6, 9, 12, 15, 18 and 21 hpi; 1, 2, 4, 5, 7, 9 and 14 dpi	Increase at 9 dpi	Blood (serum levels)	[101]
TNF	Iowa	Virulent	2 × 10^7^ TCID_50_, IN route	1, 3, 5, 7 dpi	Increase, mainly in peribronchial area	Lung	[107]
TNF	Cooper	Virulent	10^6.3^ PFU; IN route	6 dpi	Increase	Tracheal epitelium	[133]
TNF	Cooper	Virulent	10^6.3^ PFU; IN route	6 dpi	None	Retropharyngeal lymph nodes	[133]
TNF	Cooper	virulent	10^3^ PFU; IN route	20 dpi, after dexamethasone reactivation	Decrease	Tracheal epithelium	[133]
TNF	Cooper	virulent	10^3^ PFU; IN route	20 dpi, after dexamethasone reactivation	Decrease	Retropharyngeal lymph nodes	[133]
TNF	Cooper	Virulent	10^6.3^ PFU; IN route	6 dpi	Increase in frontal and posterior cortex	Nervous system	[134]
TNF	Cooper	Virulent	10^3^ PFU; IN route	24 dpi	None, little decrease in cervical medulla	Nervous system	[134]
TNF	Cooper	Virulent	10^3^ PFU; IN route	25 dpi, after dexamethasone reactivation	Increase in posterior cortex	Nervous system	[134]

hpi: hours post infection; dpi: days post infection.

**Table 3 vaccines-11-00785-t003:** BoHV-1 impact on levels of anti-inflammatory (IL-10), pro-Th1 (IL-12), and pro-Th2 (IL-4) bovine cytokines in vivo.

Cytokine	IBR Strain	Virulence of the Strain	Dose and Route of Infection	Post- Infection Time Analyzed	Impact on Cytokine’s Values	Serum/Tissue	Reference
IL-10	Iowa	Virulent	2 × 10^7^ TCID_50_, IN route	0, 3, 6, 9, 12, 15, 18, and 21 hpi; 1, 2, 4, 5, 7, 9, and 14 dpi	Increase	Blood (serum levels)	[101]
IL-10	Iowa	Virulent	2 × 10^7^ TCID_50_, IN route	1, 2, 4, 7, and 14 dpi	Increase in lymphocytes IL-10^+^, decrease in interstitial macrophage IL-10^+^	Lung	[106]
IL-4	Iowa	Virulent	2 × 10^7^ TCID_50_, IN route	0, 3, 6, 9, 12, 15, 18, and 21 hpi; 1, 2, 4, 5, 7, 9, and 14 dpi	None	Blood (serum levels)	[101]
IL-12	Iowa	Virulent	2 × 10^7^ TCID_50_, IN route	0, 3, 6, 9, 12, 15, 18, and 21 hpi; 1, 2, 4, 5, 7, 9, and 14 dpi	Increase	Blood (serum levels)	[101]
IL-12	Iowa	Virulent	2 × 10^7^ TCID_50_, IN route	1, 2, 4, 7, and 14 dpi	Increase in interstitial macrophage IL-12^+^	Lung	[106]

hpi: hours post infection; dpi: days post infection.

**Table 4 vaccines-11-00785-t004:** Modulation of circulating and tissues levels of type I and II IFNs in bovine infected with BoHV-1.

Cytokine	IBR Strain/Live Attenuated Vaccines	Virulence of the Strain	Dose and Route of Infection	Post-Infection Time Analyzed	Impact on Cytokine’s Values	Serum/ Tissue	Reference
IFN-α	108	Virulent	5 × 10^7^ PFU; IN route	0, 3, 5, 7, 10 dpi	Increase	Nasal secretion	[142]
IFN-α	108	Virulent	5 × 10^7^ PFU; IN route	0, 3, 5, 7, 10 dpi	Increase	Nasal turbinates	[142]
IFN-β	IBRV HB06	Virulent	4 × 10^5^ or 4 × 10^6^ or 4 × 10^7^ PFU; IN route	1, 2, 3, 5, 7 dpi	Increase	Blood (serum levels)	[102]
IFN-β	BoHV-1 gG−/tk−	Attenuated (HB06 derived mutant)	4 × 10^5^ or 4 × 10^6^ or 4 × 10^7^ PFU; IN route	1, 2, 3, 5, 7 dpi	Increase	Blood (serum levels)	[102]
IFN-β	BoHV-1 gG−/tk−	Attenuated (IBRV Derived mutant)	4 × 10^5^ or 4 × 10^6^ or 4 × 10^7^ PFU; IN route	1, 2, 3, 5, 7 dpi	Increase	Blood (serum levels)	[102]
IFN-β	108	Virulent	5 × 10^7^ PFU; IN route	0, 3, 5, 7, 10 dpi	Small increase 5 dpi	Nasal secretion	[142]
IFN-β	108	Virulent	5 × 10^7^ PFU; IN route	0, 3, 5, 7, 10 dpi	Increase	Nasal turbinates	[142]
IFN-β	Cooper	Virulent	10^6.3^ PFU; IN route	6 dpi	Increase, Especially nasal epithelium	Respiratory tract	[133]
IFN-β	Cooper	Virulent	10^6.3^ PFU; IN route	6 dpi	None	Retropharyngeal lymph nodes	[133]
IFN-β	Cooper	Virulent	10^3^ PFU; IN route	20 dpi, after dexamethasone reactivation	None	Respiratory tract	[133]
IFN-β	Cooper	Virulent	10^3^ PFU; IN route	20 dpi, after dexamethasone reactivation	None	Retropharyngeal lymph nodes	[133]
IFN-β	Cooper	Virulent	10^6.3^ PFU; IN route	6 dpi	Increase in cervical medulla	Nervous system	[134]
IFN-β	Cooper	Virulent	10^3^ PFU; IN route	24 dpi	Increase in cervical medulla	Nervous system	[134]
IFN-β	Cooper	Virulent	10^3^ PFU; IN route	25 dpi, after dexamethasone reactivation	Increase in cervical medulla and posterior cortex	Nervous system	[134]
IFN-γ	IBRV HB06	Virulent	4 × 10^5^ or 4 × 10^6^ or 4 × 10^7^ PFU; IN route	1, 2, 3, 5, 7 dpi	Increase	Blood (serum levels)	[102]
IFN-γ	BoHV-1 gG−/tk−	Attenuated (IBRV HB06 Derived mutant)	4 × 10^5^ or 4 × 10^6^ or 4 × 10^7^ PFU; IN route	1, 2, 3, 5, 7 dpi	Increase	Blood (serum levels)	[102]
IFN-γ	BoHV-1 gE−/tk−	Attenuated (IBRV HB06 derived mutant)	4 × 10^5^ or 4 × 10^6^ or 4 × 10^7^ PFU; IN route	1, 2, 3, 5, 7 dpi	Increase	Blood (serum levels)	[102]
IFN-γ	Iowa	Virulent	2 × 10^7^ TCID_50_, IN route	0, 3, 6, 9, 12, 15, 18 and 21 hpi; 1, 2, 4, 5, 7, 9 and 14 dpi	Increase	Blood (serum levels)	[101]
IFN-γ	BoHV-1 tmv	Attenuated (Cooper derived mutant)	2 × 10^7^ PFU, IN route	0, 7, 14, 21, 28, 35, 43 dpi; *	Increase	Blood (serum levels)	[115]
IFN-γ	BoHV-1 gEdel	Attenuated (Cooper derived mutant)	2 × 10^7^ PFU, IN route	0, 7, 14, 21, 28, 35, 43 dpi; *	Increase	Blood (serum levels)	[115]
IFN-γ	Iowa	Virulent	10^7^ TCID_50_, IN route	1, 2, 4, 7, and 14 dpi	Increase in lymphocytes IL-12+	Lung	[106]
IFN-γ	108	Virulent	5 × 10^7^ PFU; IN route	0, 3, 5, 7, 10 dpi	Increase	Nasal secretion	[142]
IFN-γ	108	Virulent	5 × 10^7^ PFU; IN route	0, 3, 5, 7, 10 dpi	None	Nasal turbinates	[142]

hpi: hours post infection; dpi: days post infection; *: challenge at 28 dpi with Cooper strain (4 × 10^7^ PFU; IN route).

## Data Availability

Not applicable.

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
