# Peer review of "The Cell-Mediated Immune Response against Bovine alphaherpesvirus 1 (BoHV-1) Infection and Vaccination"

_vaccines, 2023, doi:10.3390/vaccines11040785_

Round 1

Reviewer 1 Report

This a review manuscript. The manuscript is well written  and recommend that it should be published as written. 

Author Response

                                                                                                                      March 21th, 2023

Ms. Hazel Li

Assistant Editor

Vaccines

Dear Editor:

I wish to re-submit the manuscript titled “Evaluation of an Immunization Protocol using Bovine Alphaherpesvirus 1 gE-Deleted Marker Vaccines Against Bubaline Alphaherpesvirus 1 in Water Buffaloes.” The manuscript ID is vaccines-2304694.

We thank you for your thoughtful suggestions and insights. The manuscript has benefited from these insightful suggestions. I look forward to working with you and the reviewers to move this manuscript closer to publication in Vaccines.

The manuscript has been rechecked and the necessary changes have been made in accordance with your suggestions (Annex 1).

Thank you for your consideration. I look forward to hearing from you.

Sincerely,

Stefano Petrini, DVM, PhD, MSc

Head of National Reference Centre for Infectious Bovine Rhinotracheitis (IBR)

Istituto Zooprofilattico Sperimentale Umbria-Marche “Togo Rosati”

Via Gaetano Salvemini, 1

06126 Perugia (PG)

Italy

Tel. +39 (0)75-3433069

Fax.: +39 (0)75-35047

  1. mail: [email protected]

Annex 1

Editorial office

We notice the similarity rate is a little high in your manuscript. Please find the attached pdf with highlights and reduce similarity rate from 31% to 25% below during revisions.

- Dear Editorial office, thank you for your suggestions. We have reduced the similarity rate during revisions.

Review 2

The review entitled "The cell-mediate immune response against Bovine alphaherpesvirus 1 (BoHV-1) infection and vaccination" by Righi et al. is a comprehensive review describing the viral life cycle of BoHV-1 and the immune response to infection and vaccination. The manuscript thoroughly describes the global impact of BoHV-1 as well as the vaccines that are administered to prevent BoHV-1-associated diseases in cattle. In addition, the authors outline the innate and cellular immune responses to infection and immune processes effected by specific viral proteins. In general, the review provides molecular insight into critical immune factors and processes that control BoHV-1 infection and dissemination.

There are some issues with the text that will need to be addressed: some awkward phases on lines: 14, 15 'thanks', 43, 60, 71 and edit one sentence paragraphs.

- Dear Review, thank you for your suggestions. The sentences you suggested were modified.

Review 3

Righi et al. wrote a very interesting review on the cell-mediated immune response induced by Bovine alphaherpesvirus 1 (BoHV-1) infection or vaccination. The review is very comprehenive and accessible even for those not familiar with bovine immunology or BoHV-1. While the structure of the review is overall appropriate, and the manuscript understandable, care should be taken to perform a thorough grammatical check as many sentences seems suboptimal.

Below some comments for consideration to the author in an attempt to improve the publication.

Major comments:

POINT 1: Paragraph 5.3 – is the most relevant to the topic of the review. The authors should expand on what is known to the specific roles of CD4 and gdT cells to prevent re-activation. In case of lack of literature data on these topics, the authors should identified this as a gap and area of future research need. In addition, the authors should expand on their hypothesis related to the role of CD8 T cells which seems a bit counterintuitive from what is currently described. On one hand, the authors describe inhibition of CTLs and latency in sites that do

not express MHC class I, and on the other hand state that CD8 T cells control latency and play a crucial role in preventing reactivation.

- Dear Review, thank you for your suggestions. The paragraph was modified, with a better description of BoHV-1 protein expression during latency. During latency, viral gene expression and detection of infectious virus are extinguished, whereas genes encoded by the BoHV-1 region containing the latency-related transcript (LRT) are expressed and induce an anti-apoptotic state of the latently infected cells. Studies on HSV suggest that cytotoxic T cells might play a crucial role in preventing reactivation from latency in sensory neurons also for BoHV-1, killing infected cells as soon as BoHV-1 reactivates. The specific roles of other T cells subsets, such as CD4+ or γδ T cells, to prevent BoHV-1 re-activation from latency were not characterized in detailed, and we agree that it could be an area of future studies. Text was modified at line 408-439.

POINT 2: Paragraph 5.4 – the authors have structured the paragraph based on the assays used to assess the immune response in vaccinated calves. I would favor a structure based on the type of immune response induced and how it is linked to protection.

- Dear Review, thank you for your suggestions. In this paragraph, we described several methods to measure Th1 response. The test was modified at line 469-471 and we added a table (Table 1), where for each study we reported the assay, information on T cell response evoked, and whether it was linked to protection.

POINT 3: Paragraph 5.4 – the authors should specify if any of the vaccines are efficient to prevent latency and reactivation or only primary acute disease – and discuss the potential biomarkers involved in those different protection

- Dear Review, thank you for your suggestions. In the revised version of the manuscript, we added a table (Table 1), where we reported for each study mentioned in paragraph 5.4 the assay used, information on T cell response evoked, whether it was linked to protection prevent latency, and reactivation or only primary acute disease.

POINT 4: Paragraph 6.3 – some statements seem contradicting. It is first stated that in bovine, IL-12 does not stimulate Th1 cells, and later that “BoHV-1 infection elicits synthesis and release of IL-12, which promotes the development of a protective Th1 response”. Please clarify.

- Dear Review, thank you for your suggestions. We apologize for being unclear. In section 5.1, we did not mean IL-12 did not stimulate Th1 cells.  However, in bovine there was not a selective ability of IL-12 to stimulate these cells. We revised this sentence to avoid confusion; line 327-330.

Minor comment:

The last sentence on line 194-197 seems out of place. The authors should clarify the link between the status of BoHV-1 free and vaccination.

- Dear Review, thank you for your suggestions. We have modified the sentence as follow: “However, the use of vaccination in the European country has led to Austria, Germany, Denmark, Finland, Sweden, a region in the United Kingdom (Jersey), an area of Italy known as Valle d'Aosta and an Italian Province named Bolzano, and the Czech Republic being considered BoHV-1 free [2,54].” 

Line 200 – it is likely that both innate and adaptive immune responses are required to overcome infections. Suggest rewording to “both innate and adaptive immune responses”

- Dear Review, thank you for your suggestions. We have modified the sentence as you suggested.

Line 323-326 – sentence unclear; consider expanding as it seems to be a unique particularity of bovine Th cells that is not easily understood by non-expert

- Dear Review, thank you for your suggestions. We apologize whether we were unclear, the sentence was modified. The Th1/Th2 paradigm has provided a useful framework for understanding the observed bias in the immune response which is often dominated by cell-mediated immunity or humoral responses. Furthermore, the aforementioned paradigm has provided therapeutic strategies aimed at stimulating T cell- or antibody-mediated immunity. However, this paradigm is an oversimplification of a more complex immunoregulatory network. As reviewed by Brown et al. (1998), Th1 cells and Th2 cells represent the poles of a spectral response. In addition, the reciprocal regulation by type 1 and type 2 cytokines described in mice is not strictly conserved when human and bovine Th cells are examined. For example, in mice IL-10 is considered a type 2 cytokine, expressed by Th2 and not Th1 cells, and active only against Th1 cells. On the contrary, the IL-10 response in humans and cattle is much less restricted, and is produced by and active against all subtypes of Th cells. Text was modified at line 323-330.

Line 412 – the text seems inconsistent; it is written that viral proteins are not expressed, but the authors mentioned in previous and next sentences that LR proteins are involved in inhibiting CTL activities. Please clarify.

- Dear Review, thank you for your suggestions. We apologize for the mistake. That sentence was modified to read: During latency, viral gene expression and detection of infectious virus are extinguished, whereas genes encoded by the BoHV-1 region containing the latency-related transcript (LRT) are expressed and induce an anti-apoptotic state of the latently infected cells. The paragraph was revised (Line 408-439).

Paragraph 5.4 – no need to expand on the type of vaccines that are available here, as it is redundant with the information provided in paragraph 3

- Dear Review, thank you for your suggestions. We have deleted this part of the text.

Editorial comment:

As mentioned above, a grammatical check should be performed to improve readability (e.g. Line 24, Line 80, Line 81, Line 450, 555, 662 etc…)

- Dear Editorial office, we have done a grammar check.

Line 151 “PT” should be defined

- Dear Editorial office, we have defined a “PT”. We collected pharyngeal tonsil. 

Line 453/454 – Group A and Group B is not informative to the reader and can be deleted

- Dear Editorial office, we have deleted groups A and B.

Line 539/540 and 691/692 – it seems notes from internal review remain in the manuscript (e.g. just show the reference) – please implement.

- Dear Editorial office, we have deleted lines 539/540 and modified lines 691/692.

Line 724 and 726 – should specify “prophylaxis” in addition to treatment

- Dear Editorial office, we have modified the lines 724 and 726.

Line 727/728 – sentence seems out of place as such. It should be used to introduce the need for more research to better understand immune response to BoHV-1 and develop effective prophylaxes or treatments.

- Dear Editorial office, we have deleted the lines 727/728.

If the reviewers or editor recommended English language editing, this can be arranged by MDPI. Note that language editing by MDPI is not compulsory, nor does it guarantee that your manuscript will eventually be accepted for publication. Click on the link for more information and to request a quotation.

- Dear Editorial office, we have extensively edited the manuscript and has been significantly improved.

Reviewer 2 Report

The review entitled "The cell-mediate immune response against Bovine alphaherpesvirus 1 (BoHV-1) infection and vaccination" by Righi et al. is a comprehensive review describing the viral life cycle of BoHV-1 and the immune response to infection and vaccination. The manuscript thoroughly describes the global impact of BoHV-1 as well as the vaccines that are administered to prevent BoHV-1-associated diseases in cattle. In addition, the authors outline the innate and cellular immune responses to infection and immune processes effected by specific viral proteins. In general, the review provides molecular insight into critical immune factors and processes that control BoHV-1 infection and dissemination.

There are some issues with the text that will need to be addressed: some awkward phases on lines: 14, 15 'thanks', 43, 60, 71 and edit one sentence paragraphs.

Author Response

(The authors gave the same response as above.)

Reviewer 3 Report

Righi et al. wrote a very interesting review on the cell-mediated immune response induced by Bovine alphaherpesvirus 1 (BoHV-1) infection or vaccination. The review is very comprehenive and accessible even for those not familiar with bovine immunology or BoHV-1. While the structure of the review is overall appropriate, and the manuscript understandable, care should be taken to perform a thorough grammatical check as many sentences seems suboptimal.

Below some comments for consideration to the author in an attempt to improve the publication.

Major comments:

Paragraph 5.3 – is the most relevant to the topic of the review. The authors should expand on what is known to the specific roles of CD4 and gdT cells to prevent re-activation. In case of lack of literature data on these topics, the authors should identified this as a gap and area of future research need. In addition, the authors should expand on their hypothesis related to the role of CD8 T cells which seems a bit counterintuitive from what is currently described. On one hand, the authors describe inhibition of CTLs and latency in sites that do not express MHC class I, and on the other hand state that CD8 Tcells control latency and play a crucial role in preventing reactivation.

Paragraph 5.4 – the authors have structured the paragraph based on the assays used to assess the immune response in vaccinated calves. I would favor a structure based on the type of immune response induced and how it is linked to protection.

Paragraph 5.4 – the authors should specify if any of the vaccines are efficient to prevent latency and reactivation or only primary acute disease – and discuss the potential biomarkers involved in those different protection

Paragraph 6.3 – some statements seem contradicting. It is first stated that in bovine, IL-12 does not stimulate Th1 cells, and later that “BoHV-1 infection elicits synthesis and release of IL-12, which promotes the development of a protective Th1 response”. Please clarify.

Minor comment:

The last sentence on line 194-197 seems out of place. The authors should clarify the link between the status of BoHV-1 free and vaccination.

Line 200 – it is likely that both innate and adaptive immune responses are required to overcome infections. Suggest rewording to “both innate and adaptive immune responses”

Line 323-326 – sentence unclear; consider expanding as it seems to be a unique particularity of bovine Th cells that is not easily understood by non-expert

Line 412 – the text seems inconsistent; it is written that viral proteins are not expressed, but the authors mentioned in previous and next sentences that LR proteins are involved in inhibiting CTL activities. Please clarify.

Paragraph 5.4 – no need to expand on the type of vaccines that are available here, as it is redundant with the information provided in paragraph 3

Editorial comment:

As mentioned above, a grammatical check should be performed to improve readability (e.g. Line 24, Line 80, Line 81, Line 450, 555, 662 etc…)

Line 151 “PT” should be defined

Line 453/454 – Group A and Group B is not informative to the reader and can be deleted

Line 539/540 and 691/692 – it seems notes from internal review remain in the manuscript (e.g. just show the reference) – please implement.

Line 724 and 726 – should specify “prophylaxis” in addition to treatment

Line 727/728 – sentence seems out of place as such. It should be used to introduce the need for more research to better understand immune response to BoHV-1 and develop effective prophylaxes or treatments.

Author Response

(The authors gave the same response as above.)

Author Response

(The authors gave the same response as above.)
